# Immunohistochemical Profiling of IDO1 and IL4I1 in Head and Neck Squamous Cell Carcinoma: Interplay for Metabolic Reprogramming?

**DOI:** 10.3390/ijms26083719

**Published:** 2025-04-15

**Authors:** Benedikt Schmidl, Maren Lauterbach, Fabian Stögbauer, Carolin Mogler, Julika Ribbat-Idel, Sven Perner, Barbara Wollenberg

**Affiliations:** 1Department of Otolaryngology Head and Neck Surgery, Technical University Munich, 81675 Munich, Germany; marenwallesch@gmx.de (M.L.); barbara.wollenberg@tum.de (B.W.); 2Institute of General and Surgical Pathology, TUM School of Medicine and Health, Technical University of Munich, 81675 Munich, Germanycarolin.mogler@tum.de (C.M.); 3Department of Pathology, University of Lübeck, 23562 Lübeck, Germanysven.perner1972@googlemail.com (S.P.)

**Keywords:** IDO1, IL4I1, immunohistochemistry, TMA, metabolic reprogramming

## Abstract

Head and neck squamous cell carcinoma (HNSCC) is a heterogeneous and malignant disease with a limited number of biomarkers and insufficient targeted therapies. The current therapeutic landscape is challenged by low response rates, underscoring the need for new therapeutic targets. The success of immunotherapy in HNSCC has highlighted the importance of the immune microenvironment, and since metabolic reprogramming, especially altered tryptophan metabolism, is an important aspect in immune evasion, the interplay of the two enzymes IDO1 and IL4I1 was investigated in HNSCC to assess their immunosuppressive roles and potential as prognostic biomarkers. The immunohistochemical expression of IDO1 and IL4I1 was evaluated by an experienced head and neck pathologist in a tissue microarray (TMA) of 402 patients with HNSCC. Clinical and pathological data were retrieved, and the overall survival of the patients was calculated. In this study, IDO1 and IL4I1 were expressed by HNSCC tumor cells in the TMA of 402 patients. The overall survival analysis of the clinical data of the patients revealed that high IL4I1 expression was significantly associated with worse OS (*p* = 0.0073), while IDO1 expression did not reach statistical significance (*p* = 0.087). The combination of both markers led to a clinically significant stratification of patients. Especially p16-negative OPSCC with a high IL4I1 expression demonstrated poor survival. Immunologic differences between IDO1 and IL4I1 were detected in a TMA of 403 patients, with IDO1 and IL4I1 being expressed by HNSCC. A low IL4I1 expression in HNSCC led to a significantly better OS in this study, while IDO1 expression did not have a significant effect. Additional studies are necessary to investigate the complex interplay in the metabolic reprogramming of tumor cells.

## 1. Introduction

Head and neck squamous cell carcinoma (HNSCC) is a heterogeneous and malignant disease that originates from the mucosa of the upper aerodigestive tract and is associated with a poor prognosis and a lack of preoperative and prognostic biomarkers [1]. The heterogeneity of the disease impairs the use of targeted therapy and is further complicated by an insufficient knowledge of the interplay of the immune system and tumor cells of the tumor microenvironment (TME) [2]. The currently approved immunotherapy agents for HNSCC target the PD-1 receptor on lymphocytes and thereby block ligands that could deactivate them [3]. PD-1 expression is an important mechanism contributing to the exhausted effector T cell phenotype and the expression of PD-1 on effector T cells, and PD-L1 on neoplastic cells enables tumor cells to evade anti-tumor immunity [4]. Currently, this treatment is approved for the recurrent/metastatic HNSCC using an IgG4 humanized antibody against programmed cell death 1 (PD1). While the introduction of PD-1 inhibition has revolutionized the therapeutic landscape, there is still a response rate of only 20% for HNSCC [5,6]. For that reason, there is a need to investigate novel targets for (immune-)therapy that can be used to develop drugs either as a standalone therapy or in combination with existing immunotherapy agents [7,8].

Since metabolic reprogramming is one of the most prominent features of HNSCC, with amino acid metabolism as the most significantly altered one, elucidating aberrant metabolic profiles might be the key to understanding the mechanisms of tumor immune escape [9].

One of these potential candidates is therefore indoleamine 2, 3-dioxygenase 1 (IDO/IDO1/INDO), a rate-limiting enzyme that metabolizes the essential amino acid, tryptophan (Trp), into downstream kynurenines (Kyn) [10]. IDO1 has a potential immunosuppressive role with IDO and/or tryptophan dioxygenase (TDO)-mediated depletion of Trp and/or the accumulation of Kyn, which is associated with the suppression of immune effector cells and the upregulation, activation, and/or induction of tolerogenic immune cells [11,12]. A high expression of IDO1 leads to a high rate of tryptophan conversion and depletion. This induces cell cycle arrest and/or anergy in the effector cytotoxic lymphocyte (CTL) compartment. This also leads to the activation/maturation of regulatory T cells (Treg) in association with CTLA4-mediated CD80/CD86 co-inhibition. Kynurenine (Kyn) also directly induces the apoptosis of CTL. TGF-β signaling results in the phosphorylation of the IDO1 intrinsic immunoreceptor tyrosine-based inhibitory motifs (ITIM), leading to non-canonical NF-κB activation and autocrine reinforcement of IDO1 and TGF-β expression [12,13,14]. There is a growing number of clinical trials focused on IDO1, with many studies coupling multiple substances to test the combinatorial benefit [12,15].

Another promising target in the metabolic landscape of HNSCC is interleukin 4-induced gene 1 (IL4I1), an amino acid-catabolizing enzyme, that is mainly secreted in the synaptic cleft and expressed by antigen-presenting cells ([3]). As IDO1, IL4I1 also generates bioactive metabolites from tryptophan. In scRNAseq data, there is an overlapping expression pattern in myeloid cells of the tumor microenvironment, suggesting the two enzymes control a network of tryptophan-specific metabolic events [16]. IL4I1 activates the aryl hydrocarbon receptor through the generation of indole metabolites and kynurenic acid and is associated with reduced survival in glioma patients. IL4I1 can suppress T cell proliferation and promote cancer cell motility and suppress adaptive immunity in chronic lymphocytic leukemia (CLL) in mice. It has been suggested that since IDO1 inhibitors do not block IL4I1, IL4I1 may be the reason for the prior failure of clinical studies combining immunotherapy with IDO1 inhibition [17].

Only a few studies have investigated the role of IDO1 and IL4I1 in HNSCC so far, with limited insights into the interplay of IDO1 and IL4l1, but they have demonstrated that IDO1 expression in HNSCC is positively correlated with several immune-related molecules, and that IL4I1 is a metabolic immune checkpoint that activates the aryl hydrocarbon receptor (AHR) and promotes tumor progression [18,19]. The objective of this study was therefore to evaluate the expression of IDO and IL4I1 in a tissue microarray of HNSCC patients to assess a functional and prognostic role and to discover the dynamic immune landscape within HNSCC.

## 2. Results

For the analysis, the TMA cores were evaluated by an experienced head and neck pathologist and the mean expression of IDO1 and IL4I1 in the tumor was calculated and used for the subsequent analysis. The TMA cohort had an even distribution of early (52.8%) and advanced HNSCC (47.2%). Distant metastasis was present in 12.9% and regional lymph node metastasis in 37.9% of the cases. There was an even distribution of anatomical sites, with 26.0% oropharyngeal squamous cell carcinoma (OPSCC) and 29.5% laryngeal carcinoma as the most common sites. A total of 53.1% of the 128 OPSCC demonstrated a positive p16 staining (68 vs. 60). A total of 87.9% of the patients were smokers (313 out of 356). Regular alcohol consumption was reported in 43.0% of the patients. The overall clinical and pathological characteristics are depicted in Figure 1.

### 2.1. The Expression of IDO1 and IL4I1 and Association with Clinicopathologic Characteristics

Both IDO1 and IL4I1 were expressed by HNSCC tumor cells in the TMA. Exemplary images of the immunohistochemical staining of IDO1 and IL4I1 are depicted in Appendix A.

To assess the optimal cutoff of the expression of IDO1 and IL4I1, maximally selected rank statistics were calculated using the “maxstat method” in R. To differentiate high and low expression groups, a cutoff of 14% of IDO1 positivity was calculated, while for IL4I1 any expression was considered positive/high expression. For IDO1, 68 patients (19.1%) were classified as low expression, while 288 patients (80.9%) were classified as high expression. The mean expression for IDO1 was 10.5%, with a standard deviation of 8.5%, reflecting variability primarily due to the wide range of expression levels in the high-expression group. For IL4I1, 236 patients (66.3%) were classified as low expression, and 120 patients (33.7%) were classified as high expression. The mean expression for IL4I1 was 2.3%, with a standard deviation of 3.0%, indicating that while most samples had low expression, there was still considerable variation in the high-expression group (Appendix A).

The associations were tested only in patients with complete clinical data, resulting in a cohort of 356 patients. The comparison of the clinical data and the immunohistochemical expression using Fisher’s exact test revealed no association of IDO1 expression with T-stage (*p* = 0.058), nor with lymphatic metastasis (*p* = 0.091). When looking at the different anatomical locations of the HNSCC, the majority of hypopharyngeal carcinoma showed a high expression of IDO1, as well as p16-negative and p16-positive oropharyngeal squamous cell carcinoma (OPSCC). Distant metastasis, gender, p16 expression in general, and recurrence did not have significant associations, whereas smokers were more often in the high-expression group (*p* = 0.023) (Figure 2).

For IL4I1, there was a significant association with lymphatic metastasis (*p* = 0.021), while T-stage, M-stage, p16 expression, smoking, gender, recurrence, and alcohol consumption were not associated significantly (Figure 2). Hypopharyngeal carcinoma, p16-negative OPSCC, and p16-positive OPSCC demonstrated low Il4I1 expressions more often.

### 2.2. Prognostic Impact of the Expression of IDO1 and IL4I1

Next, the clinical data of the 402 patients in the TMA were used to calculate Kaplan–Meier survival curves and log-rank testing was applied to calculate statistical significance. A high expression of IDO1 in the cohort of all patients was associated with poor OS, but did not reach significance (*p* = 0.087), while in the different subgroups stratified by the anatomical location, there was a no significantly improved survival in the different subgroups. Since the expression of p16 is the most important prognostic factor in OPSCC, the cases were also stratified as p16-positive or p16-negative, but there was no statistically significant differentiation of survival based on IDO1 expression.

At the same time, the expression of IL4I1 in the cohort of all patients was associated with a significantly worse overall survival (*p* = 0.0073), whereas in the different subgroups stratified by anatomical location, there was a significantly worse OS in the group of p16-negative OPSCC (*p* = 0.018). The other subgroups did not achieve a significant patient stratification.

In the next step, IDO1 expression and IL4I1 expression were combined in univariate survival analysis, resulting in a significant stratification of patients with the group of high IDO1, and high IL4I1 expression resulted in the worst overall survival, since the group of low IDO1 expression and high IL4I only involved 17 patients. The group of low IDO1 and low IL4I1 expression had the best overall survival of the cohort (Figure 3). 

In the cohort of 356 patients, multivariable Cox proportional hazards regression analysis demonstrated that a lower expression of IDO1 (hazard ratio = 1.4278, *p* = 0.3286) and IL4I1 (hazard ratio = 0.7953, *p* = 0.2264) did not significantly reduce the risk of events individually. However, the interaction between low IDO and low IL4I1 expression achieved a slightly significant decrease in hazard (hazard ratio = 0.3301, *p* = 0.0263).

## 3. Discussion

IDO1 has been presented as a novel immune-related gene in oral squamous cell carcinoma (OSCC) in the mRNA sequencing data of the TCGA dataset [20], potentially associated with tumor progression, immune evasion, and suppression, and IDO1 inhibitors have already been used in clinical trials for cancer immunotherapy in other carcinomas [21]. At the same time, some of the first clinical trials have already failed in highly immunogenic tumors such as melanoma [22]. There are data that suggest that immune checkpoint blockade (ICB) might induce the expression of IL4I1, an amino acid-catabolizing enzyme that is also involved in metabolic reprogramming, and thereby compromise the effect of IDO inhibition [17].

The objective of this study was therefore to investigate the controversial role of IDO1 and IL4I1 in the largest cohort so far of HNSCC to lay the foundation for a potential use of a combined therapeutic approach in the future. The analysis of the expression of IDO and IL4I1 in this study revealed no significant impact of the expression of IDO1 on the overall survival of patients, while there was a significant association of low expression of IL4I1 leading to a better survival. Both markers were expressed by HNSCC cells in the TMA, and the expression of IDO1 was associated with smoking, whereas tumors with low expression of IL4I1 had less lymphatic metastasis. When analyzing the subgroups stratified by anatomical localization, there is a significantly worse survival of p16-negative OPSCC with a high IL4I1 expression. When both IDO1 and IL4I1 were combined, the survival of the low IDO1 and low IL4I1 was significantly worse than the other groups.

The results for IDO1 in this study elaborate the findings of a bioinformatic study showing a significantly higher expression of IDO1 in HNSCC in the TCGA dataset, especially in HPV + HNSCC compared to healthy control tissue [18], but no prognostic role for overall and disease-free survival [18]. Another study did not focus on the IDO1 expression in tumor cells, but found IDO1 expressing immune cells, especially macrophages, to be more abundant in advanced stages of OSCC and a reduced progression-free survival [19]. Simultaneously, there are studies in other tumors showing that a high IDO1 expression might reflect the presence of a T cell-inflamed phenotype, associated with good prognosis and potentially a response to immunotherapy and chemotherapy [23]. IDO1 was even proposed as a surrogate marker for a more robust spontaneous antitumor immune response in some cancers [24], backed by studies evaluating the IDO1 expression in circulating tumor cells (CTCs) at baseline and after completion of chemoradiotherapy, the finding of a significant overexpression at baseline compared with the post-treatment counterparts, and that mRNA expression at baseline is associated with better survival in terms of progression-free survival. Controversially, in the same study, post-treatment IDO1 mRNA levels were correlated with unfavorable prognosis in terms of overall survival [23].

For Il4I1 there is only a single study so far that investigated the role of IL4I1 in head–neck cancer-derived mesenchymal stromal cells (MSC) in the microenvironment. A microarray gene revealed that HNSCC-MSCs in response to IFN-γ and TNF-α express IL4I1, which is then able to suppress T cell proliferation in vitro [25]. A clinical neoadjuvant study of pembrolizumab for oral tongue squamous cell carcinoma found IL4I1 as a differentially expressed fatty acid metabolism-related gene and a high expression in anti-PD1 therapy responders [26]. Unfortunately, there was no clinical correlation and validation of this bioinformatic result, which might have been able to back the theory of increased IL4I1 expression in patients with anti-PD1 therapy [17]. The data from our study highlight the importance of IL4I1 and IDO1 expression in HNSCC and suggest a potential interplay between these enzymes, which is especially important since it was suggested that IL4I1 may be the reason for the prior failure of clinical studies combining immunotherapy with IDO1 inhibition [17]. Specifically, the combination of low IDO1 and low IL4I1 expression in this study of HNSCC tissue was associated with a significant decrease in hazard, indicating a possible synergistic effect on overall survival. While our study is based on immunohistochemical expression analysis, these findings support the hypothesis that IDO1 and IL4I1 may cooperatively influence the tumor immune microenvironment, possibly through metabolic reprogramming and immune evasion mechanisms. Although direct functional assays are necessary to elucidate this interplay further, our results provide a strong foundation for future investigations, including metabolic profiling and mechanistic studies, to investigate the precise biological interactions between these two immunosuppressive enzymes.

While the results of this study highlight the increasing importance of the interplay of IDO and IL4I1 in HNSCC, there are a few limitations. Since HNSCC are quite heterogeneous, the small tissue cores that are used to generate a TMA might not fully represent this, especially when the marker expression is unevenly distributed across the tumor tissue. This might even lead to over or underestimation of the marker expression [27,28]. Another important aspect is that the expression of IDO and IL4I1 in this study represents only a single-time-point snapshot. In the future, biopsies from recurrence, or a correlation with liquid-biopsy samples, might show an even more dynamic change in the two markers [29,30]. Additionally, the visual assessment of the IDO and IL4I1 expression relies on the interpretation of a pathologist. For this reason, scoring criteria were predefined and the pathologist was blinded. Another limitation was the lack of some of the clinical data resulting in lower patient numbers for the testing of associations, which might have resulted in the worse survival data for the relatively small group of low IDO1 and high IL4I1 expression (n = 17). Since automatic cutoff calculation was used in this study due to the novelty of the markers, clinically more meaningful cutoffs might lead to different results in the future. Additionally, the character of this study does not allow the establishment of causal relationships or mechanisms but lays the foundation for future studies investigating the inhibition of both markers in the clinical setting.

## 4. Materials and Methods

### 4.1. Patient Cohort

This study included a well-characterized cohort (Lübeck cohort) of 402 patients with an HNSCC, who were treated according to local treatment guidelines at the Department for Otorhinolaryngology of the University Hospital Schleswig-Holstein, Campus Lübeck, Germany, between 2001 and 2016. Clinical data, survival data, and patient characteristics were obtained from the Department for Otorhinolaryngology of the University Hospital Schleswig-Holstein, Campus Lübeck, Germany. To ensure patient confidentiality, the clinical data were anonymized before being shared with the researchers, making patient identification impossible. This study was approved by the ethics committee of the Technical University of Munich The characteristics of the patient cohort are depicted in Figure 1.

### 4.2. Immunohistochemistry and the Assessment of the IDO1 and IL4I1 Expression

Formalin-fixed paraffin-embedded (FFPE) tumor tissues were retrieved from the archives of the Institute of Pathology of the University Hospital Schleswig-Holstein, Campus Lübeck, Germany and tissue microarrays (TMA) were constructed from representative tumor areas. Three 0.6 mm cores of each tumor specimen were assembled into TMA blocks. The FFPE tissue was cut with a microtom into 2–3 μm thick sections and deparaffinized at 65 °C. Immunohistochemical staining was conducted with the IDO1 primary antibody (1:100, Thermo Fisher, Salem, MA, USA) and IL4I1 primary antibody (1:100, Thermo Fisher, Salem, MA, USA). Slides were counterstained with hematoxylin. After dehydration by immersion in the ethanol series and xylol (2 min each) the slides were examined under light microscopy.

The immunohistochemically stained slides were evaluated by an experienced pathologist (F.S.) using a standardized scoring protocol to ensure reproducibility and minimize observer bias. For each case, five high-power fields (HPF) were selected from representative tumor areas, and the percentage of tumor cells exhibiting positive cytoplasmic staining for IDO1 or IL4I1 was assessed separately for each HPF. The average percentage across these HPFs was calculated and used for further analysis. To determine the optimal cutoff values for high versus low expression of IDO1 and IL4I1, we employed the “maxstat” function in R, which performs maximally selected rank statistics to identify the threshold that best stratifies the data based on outcome measures [31]. The cutoff value yielding the highest Youden Index was selected to maximize sensitivity and specificity. This approach ensures an objective, data-driven threshold for classification. To account for multiple testing, *p*-values were adjusted using the Bonferroni correction method, thereby reducing the risk of false positive results. A *p*-value of less than 0.05 was considered statistically significant. The summary data of the immunohistochemistry scoring, including the distribution of high and low expression cases, mean expression levels, and correlation with clinical parameters, are provided in the results section.

### 4.3. Statistical Analysis

Kaplan–Meier survival analysis and log-rank testing were used to compare survival rates for different patient groups and clinical characteristics. Associations were tested with Fisher’s exact test and Bonferroni correction. At *p* < 0.05 the null hypothesis was rejected, and the result was considered statistically significant. Statistical calculation was performed using Prism (Version 10.4.2) (GraphPad Software, La Jolla, CA, USA) and R (Version 2.15.2, R Core Team, 2024) using the survival, ggplot, and coxph packages.

## 5. Conclusions

This study is the first highlighting the potential of a combination of two markers of the tryptophan pathway and to assess the interplay of these markers for the progression of HNSCC and the impact on overall survival. The results demonstrate a complex interplay with IL4I1 being shown as a prognostic and important factor in HNSCC, laying the foundation for further studies into analyzing the impact of dual therapy of these markers in vitro for HNSCC.

## Figures and Tables

**Figure 1 ijms-26-03719-f001:**
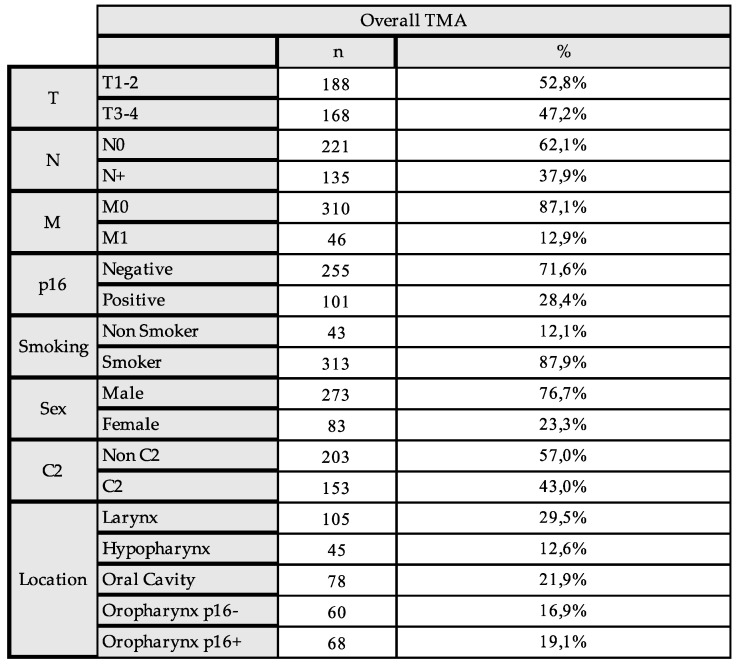
**Clinical and pathological data of the HNSCC TMA Cohort of 402 patients.** Depicted is the percentage of the total number of the category. Abbreviations: C2 = Alcohol Consumption, CUP = Cancer of Unknown Primary.

**Figure 2 ijms-26-03719-f002:**
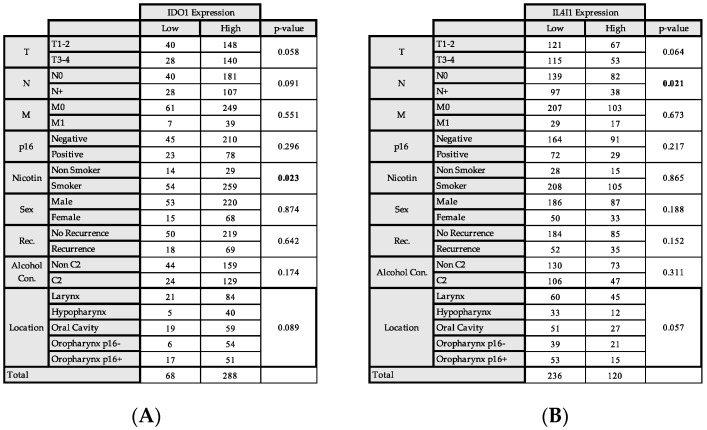
Association of IDO1 and IL4L1 Expression and clinical and pathological data (**A**) Expression of IDO1 and. (**B**) Expression of IL4L1 and association with clinical and pathological data. Fishers Exact Test was used to calculate the statistical significance. A *p*-value of 0.05 was considered significant.

**Figure 3 ijms-26-03719-f003:**
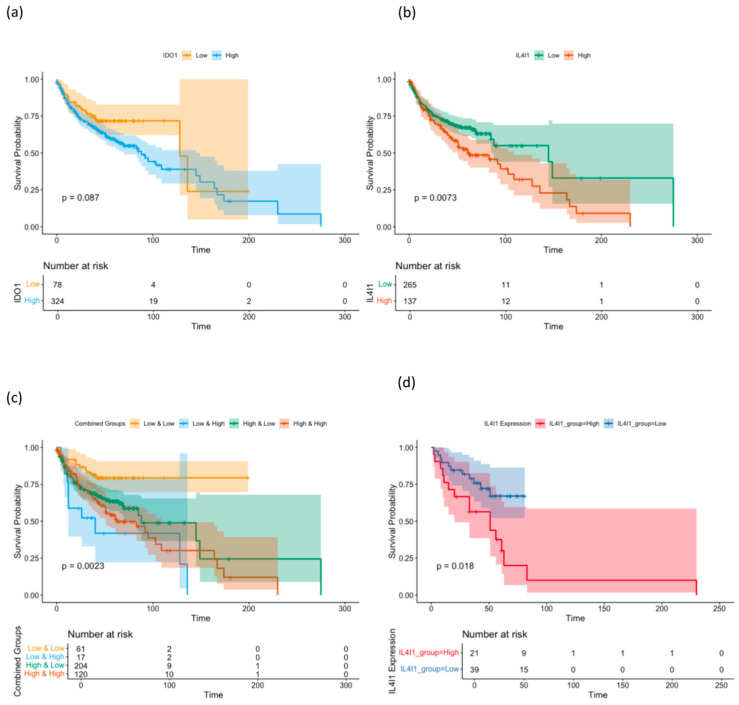
Kaplan-Meier survival curves stratified by (**a**) IDO1 and (**b**) IL4I1 and (**c**) combined expression levels in head and neck squamous cell carcinoma (HNSCC) patients. (**d**) Subanalysis of p16-negative OPSCC stratified by IL4I1 expression. Patients were divided into high and low expression groups based on the optimal cutoffs for IL4I1 and IDO1 expression. The curves illustrate survival probabilities over time and the number at risk. Statistical significance was assessed using the log-rank test.

## Data Availability

The original contributions presented in this study are included in the article/Appendix A. Further inquiries can be directed to the corresponding author(s).

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
