# Peer review of "Immunohistochemical Profiling of IDO1 and IL4I1 in Head and Neck Squamous Cell Carcinoma: Interplay for Metabolic Reprogramming?"

_ijms, 2025, doi:10.3390/ijms26083719_

Round 1
Reviewer 1 Report
Comments and Suggestions for Authors
A nice paper on the importance and interplay of IDO1 and IL4I1 in HNSCC.
In line with the cited literature, the Kaplan-Meier curves show reduced survival with high IL4I1 expression. This was also discussed coherently. However, at the beginning of the manuscript, the abstract states that higher IL4I1 expression led to better OS. It is the other way round. A correction is recommended here.
The tables in figure 2 show the p-values ​​of the Fishers Exact Test. Please indicate the significance level in the legend.
Please also check the legend for Figure 3. It seems that parts of the template still remain here. These can be deleted.
Overall, the manuscript is well written and offers an interesting perspective for new findings in the field of tumor immunology.
Author Response
Dear Reviewer, We would like to sincerely thank you for your thoughtful and constructive feedback on our manuscript.
Reviewer #1:
A nice paper on the importance and interplay of IDO1 and IL4I1 in HNSCC.
In line with the cited literature, the Kaplan-Meier curves show reduced survival with high IL4I1 expression. This was also discussed coherently. However, at the beginning of the manuscript, the abstract states that higher IL4I1 expression led to better OS. It is the other way round. A correction is recommended here.
Detailed Response to Reviewer #1:
Thank you for your careful review and constructive feedback. We have corrected the abstract to accurately reflect that higher IL4I1 expression is associated with reduced overall survival, aligning with our Kaplan-Meier analyses and discussion.
Reviewer #1:
The tables in figure 2 show the p-values ​​of the Fishers Exact Test. Please indicate the significance level in the legend. Please also check the legend for Figure 3. It seems that parts of the template still remain here. These can be deleted. Overall, the manuscript is well written and offers an interesting perspective for new findings in the field of tumor immunology.
Detailed Response to Reviewer #1:
We have updated the legend for Figure 2 to indicate the significance level of the Fisher’s Exact Test and have revised the legend for Figure 3 to remove the remaining template text. We appreciate your positive remarks on our manuscript and its contribution to tumor immunology and thank you for helping us improve its clarity and accuracy.
Reviewer #2:
The detailed information of immunohistochemistry scoring should be shown in the method and the summary data of this scoring should be shown in the results.
Detailed Response to Reviewer #2:
We acknowledge the need for a more detailed description of our IHC scoring methodology. We have now expanded the Methods section to provide a comprehensive explanation of the scoring criteria, including the evaluation process, statistics, cutoff values used to define high and low expression. The summary data is provided in the results section.
Reviewer 2 Report
Comments and Suggestions for Authors
The manuscript titled “Immunohistochemical Profiling of IDO1 and IL4I1 in Head and Neck Squamous Cell Carcinoma: Interplay for Metabolic Reprogramming?” by Schmidl et al. demonstrated that IDO1 and IL4I1 was highly expressed in HNSCC patients. The overall survival analysis revealed no significant association with the expression of IDO1 and a significantly better survival with a high expression of IL4I1. The combination of IDO1 and IL4I1 did not lead to a significant stratification of patients. There was no significant association with the expression of p16. The metabolic reprogramming is one of the most prominent features of HNSCC with the amino acid metabolism as the most significantly altered one, elucidating aberrant metabolic profiles might be the key for understanding the mechanisms of tumor immune escape. Thus, the biomedical rationale for this study is sound and interest. However, there are some comments for improvement.
Comments:
- The detailed information of immunohistochemistry scoring should be shown in the method and the summary data of this scoring should be shown in the results.
- In the immunohistochemical data, “IL4I1 High Expression” seems not with high expression of IL4I1. However, the authors should show the whole image of this case in the tissue microarrays for IDO1 and IL4I1, except for the partially enlarged images.
- Notably, in immunohistochemical images, the scale bar is the same (50 μm), but the images showed by the authors were not at the same magnification. Why?
- The expression of IDO1 or IL4I1 was already reported in the HNSCC patients. In this study, the novelty is the interplay for metabolic reprogramming, however, this part was not well performed in this study.
Author Response
Dear Reviewer,
We would like to sincerely thank you for your thoughtful and constructive feedback on our manuscript.
We have carefully addressed all the raised concerns:
Reviewer #2:
The detailed information of immunohistochemistry scoring should be shown in the method and the summary data of this scoring should be shown in the results.
Detailed Response to Reviewer #2:
We acknowledge the need for a more detailed description of our IHC scoring methodology. We have now expanded the Methods section to provide a comprehensive explanation of the scoring criteria, including the evaluation process, statistics, cutoff values used to define high and low expression. The summary data is provided in the results section and the supplementary material.
Reviewer #2:
In the immunohistochemical data, “IL4I1 High Expression” seems not with high expression of IL4I1. However, the authors should show the whole image of this case in the tissue microarrays for IDO1 and IL4I1, except for the partially enlarged images.
Notably, in immunohistochemical images, the scale bar is the same (50 μm), but the images showed by the authors were not at the same magnification. Why?
Detailed Response to Reviewer #2:
We appreciate your concern regarding the representation of IL4I1 expression. To provide a clearer and more comprehensive view, we have fully revised the figure now including whole-slide images of the tissue microarrays for both IDO1 and IL4I1 in the supplementary material. These images will allow readers to assess the overall expression pattern, complementing the previously provided enlarged images.
Reviewer #2:
The expression of IDO1 or IL4I1 was already reported in the HNSCC patients. In this study, the novelty is the interplay for metabolic reprogramming, however, this part was not well performed in this study.
Detailed Response to Reviewer #2:
We appreciate your comment and would like to clarify the unique contributions of our study. To our knowledge, this is the first study to describe IL4I1 expression in a TMA of HNSCC tumor tissue, providing novel insights into its potential role in tumor biology and immune modulation. Additionally, we analyse IDO1 expression in the largest HNSCC cohort to date, strengthening the validity of this marker in HNSCC. By analysing the expression of both markers in a large patient cohort, we highlight correlations that suggest a possible cooperative role in shaping the tumor microenvironment. To better highlight this, we have revised the Discussion section to emphasize how the combined expression of these enzymes may shape the tumor microenvironment by modulating amino acid metabolism and immune evasion. These revisions ensure that the novelty and significance of our study are more clearly articulated.
Once again, we are grateful for the opportunity to resubmit and for your thoughtful guidance throughout the review process. We look forward to your consideration of our manuscript.
Round 2
Reviewer 2 Report
Comments and Suggestions for Authors
No more comments